# Maximum Causal Tsallis Entropy Imitation Learning

**Kyungjae Lee[1], Sungjoon Choi[2], and Songhwai Oh[1]**
Dep. of Electrical and Computer Engineering and ASRI, Seoul National University[1]
Kakao Brain[2]
kyungjae.lee@rllab.snu.ac.kr, sam.choi@kakaobrain.com,
songhwai@snu.ac.kr

## Abstract

In this paper, we propose a novel maximum causal Tsallis entropy (MCTE) framework for imitation learning which can efficiently learn a sparse multi-modal policy distribution from demonstrations. We provide the full mathematical analysis of the proposed framework. First, the optimal solution of an MCTE problem is shown to be a sparsemax distribution, whose supporting set can be adjusted. The proposed method has advantages over a softmax distribution in that it can exclude unnecessary actions by assigning zero probability. Second, we prove that an MCTE problem is equivalent to robust Bayes estimation in the sense of the Brier score. Third, we propose a maximum causal Tsallis entropy imitation learning (MCTEIL) algorithm with a sparse mixture density network (sparse MDN) by modeling mixture weights using a sparsemax distribution. In particular, we show that the causal Tsallis entropy of an MDN encourages exploration and efficient mixture utilization while Shannon entropy is less effective.

## 1  Introduction

In this paper, we focus on the problem of imitating demonstrations of an expert who behaves non-deterministically depending on the situation. In imitation learning, it is often assumed that the expert's policy is deterministic. However, there are instances, especially for complex tasks, where multiple action sequences perform the same task equally well. We can model such nondeterministic behavior of an expert using a stochastic policy. For example, expert drivers normally show consistent behaviors such as keeping lane or keeping the distance from a frontal car, but sometimes they show different actions for the same situation, such as overtaking a car and turning left or right at an intersection, as suggested in [1]. Furthermore, learning multiple optimal action sequences to perform a task is desirable in terms of robustness since an agent can easily recover from failure due to unexpected events [2, 3]. In addition, a stochastic policy promotes exploration and stability during learning [4, 2, 5]. Hence, modeling experts' stochasticity can be a key factor in imitation learning.

To this end, we propose a novel maximum causal Tsallis entropy (MCTE) framework for imitation learning, which can learn from a uni-modal to multi-modal policy distribution by adjusting its supporting set. We first show that the optimal policy under the MCTE framework follows a *sparsemax* distribution [6], which has an adaptable supporting set in a discrete action space. Traditionally, the maximum causal entropy (MCE) framework [1, 7] has been proposed to model stochastic behavior in demonstrations, where the optimal policy follows a softmax distribution. However, it often assigns non-negligible probability mass to non-expert actions when the number of actions increases [3, 8]. On the contrary, as the optimal policy of the proposed method can adjust its supporting set, it can model various expert's behavior from a uni-modal distribution to a multi-modal distribution.

To apply the MCTE framework to a complex and model-free problem, we propose a maximum causal Tsallis entropy imitation learning (MCTEIL) with a sparse mixture density network (sparse MDN) whose mixture weights are modeled as a sparsemax distribution. By modeling expert's behavior

using a sparse MDN, MCTEIL can learn varying stochasticity depending on the state in a continuous action space. Furthermore, we show that the MCTEIL algorithm can be obtained by extending the MCTE framework to the generative adversarial setting, similarly to generative adversarial imitation learning (GAIL) by Ho and Ermon [9], which is based on the MCE framework. The main benefit of the generative adversarial setting is that the resulting policy distribution is more robust than that of a supervised learning method since it can learn recovery behaviors from less demonstrated regions to demonstrated regions by exploring the state-action space during training. Interestingly, we also show that the Tsallis entropy of a sparse MDN has an analytic form and is proportional to the distance between mixture means. Hence, maximizing the Tsallis entropy of a sparse MDN encourages exploration by providing bonus rewards to wide-spread mixture means and penalizing collapsed mixture means, while the causal entropy [1] of an MDN is less effective in terms of preventing the collapse of mixture means since there is no analytical form and its approximation is used in practice instead. Consequently, maximizing the Tsallis entropy of a sparse MDN has a clear benefit over the causal entropy in terms of exploration and mixture utilization.

To validate the effectiveness of the proposed method, we conduct two simulation studies. In the first simulation study, we verify that MCTEIL with a sparse MDN can successfully learn multi-modal behaviors from expert's demonstrations. A sparse MDN efficiently learns a multi-modal policy without performance loss, while a single Gaussian and a softmax-based MDN suffer from performance loss. The second simulation study is conducted using four continuous control problems in MuJoCo [10]. MCTEIL outperforms existing methods in terms of the average cumulative return. In particular, MCTEIL shows the best performance for the *reacher* problem with a smaller number of demonstrations while GAIL often fails to learn the task.

## 2 Related Work

The early researches on IRL [1, 11–18] can be categorized into two groups: a margin based and entropy based method. A margin based method maximizes the margin between the value of the expert's policy and all other policies [11, 12]. In [11], Abbeel and Ng proposed an apprenticeship learning where the rewards function is estimated to maximize the margin between the expert's policy and randomly sampled policies. In [12], Ratliff et al. proposed the maximum margin planning (MMP) where Bellman-flow constraints are introduced to consider the margin between the experts' policy and all other possible policies. On the contrary, an entropy based method is first proposed in [1] to handle the stochastic behavior of the expert. Ziebart et al. [1] proposed a maximum entropy inverse reinforcement learning (MaxEnt IRL) using the principle of maximum (Shannon) entropy to handle ambiguity issues of IRL. Ramachandran et al. [13] proposed Bayesian inverse reinforcement learning (BIRL) where the Bayesian probabilistic model over demonstrations is proposed and the expert policy and rewards are inferred by using a Metropolis-Hastings (MH) method. In[1, 13], the expert behavior is modeled as a softmax distribution of an action value which is the optimal solution of the maximum entropy problem. We also note that [14–18] are variants based on [1, 13].

In [9], Ho and Ermon have extended [1] to a unified framework for two groups by adding a reward regularization. Most existing IRL methods can be interpreted as the unified framework with different reward regularization. Those methods including the aforementioned algorithms [1, 11–18] require to solve an MDP problem every iterations to update a reward function. In model-free case, reinforcement learning (RL) method should be applied to solve the MDP, which leads to high computational costs and huge amounts of samples. To address this issue, Ho and Ermon proposed the generative adversarial imitation learning (GAIL) method where the policy function is updated to maximize the reward function and the reward function is updated to assign high values to expert's demonstrations and low values to trained policy's demonstrations. GAIL achieves sample efficiency by avoiding the need to solve RL as a subroutine and alternatively updating policy and reward functions.

Recently, several variants of GAIL [19–21] have been developed based on the maximum entropy framework. These methods [19–21] focus on handling the multi-modality in demonstrations by learning the latent structure. In [19], Hausman et al. proposed an imitation learning method to learn policies using unlabeled demonstrations collected from multiple different tasks where the latent intention is introduced in order to separate mixed demonstrations. Similarly, in [20], a robust imitation learning method is proposed, which separates unlabeled demonstrations by assigning the latent code using a variational autoencoder. The encoding network assigns the latent code to the input demonstration. Then, the policy network is trained to mimic the input demonstration given the

latent code and the encoding network is trained to recover the given latent code from the generated trajectory. In [21], the latent code is also proposed to handle multi-modal demonstrations. The latent structure in [21] is learned by maximizing the lower bound of mutual information between the latent code and the corresponding demonstrations. Consequently, existing imitation learning methods which can handle the multi-modal behavior have common features in that they are developed based on the maximum entropy framework and capture the multi-modality of demonstrations by learning the mapping from demonstrations to the latent space.

Unlikely to recent methods for multi-modal demonstrations, the proposed method is established on the maximum causal Tsallis entropy framework which induces a sparse distribution whose supporting set can be adjusted, instead of the original maximum entropy. Furthermore, a policy is modeled as a sparse mixture density network (sparse MDN) which can learn multi-modal behavior directly instead of learning the latent structure.

## 3  Background

**Markov Decision Processes**  Markov decision processes (MDPs) are a well-known mathematical framework for a sequential decision making problem. A general MDP is defined as a tuple $\{\mathcal{S}, \mathcal{F}, \mathcal{A}, \phi, \Pi, d, T, \gamma, \mathbf{r}\}$, where $\mathcal{S}$ is the state space, $\mathcal{F}$ is the corresponding feature space, $\mathcal{A}$ is the action space, $\phi$ is a feature map from $\mathcal{S} \times \mathcal{A}$ to $\mathcal{F}$, $\Pi$ is a set of stochastic policies, i.e., $\Pi = \{\pi \mid \forall s \in \mathcal{S},\ a \in \mathcal{A},\ \pi(a|s) \geq 0 \text{ and } \sum_{a'} \pi(a'|s) = 1\}$, $d(s)$ is the initial state distribution, $T(s'|s, a)$ is the transition probability from $s \in \mathcal{S}$ to $s' \in \mathcal{S}$ by taking $a \in \mathcal{A}$, $\gamma \in (0, 1)$ is a discount factor, and $\mathbf{r}$ is the reward function from a state-action pair to a real value. In general, the goal of an MDP is to find an optimal policy distribution $\pi^* \in \Pi$ which maximizes the expected discount sum of rewards, i.e., $\mathbb{E}_\pi [\mathbf{r}(s, a)] \triangleq \mathbb{E}\left[\sum_{t=0}^{\infty} \mathbf{r}(s_t, a_t)|\pi, d\right]$. Note that, for any function $f(s, a)$, $\mathbb{E}\left[\sum_{t=0}^{\infty} f(s_t, a_t)|\pi, d\right]$ will be denoted as $\mathbb{E}_\pi [f(s, a)]$.

**Maximum Causal Entropy Inverse Reinforcement Learning**  Zeibart et al. [1] proposed the maximum causal entropy framework, which is also known as maximum entropy inverse reinforcement learning (MaxEnt IRL). MaxEnt IRL maximizes the causal entropy of a policy distribution while the feature expectation of the optimized policy distribution is matched with that of expert's policy. The maximum causal entropy framework is defined as follows:

$$
\begin{aligned}
\underset{\pi \in \Pi}{\text{maximize}} \quad & \alpha H(\pi) \\
\text{subject to} \quad & \mathbb{E}_\pi \left[\phi(s, a)\right] = \mathbb{E}_{\pi_E} \left[\phi(s, a)\right],
\end{aligned}
\tag{1}
$$

where $H(\pi) \triangleq \mathbb{E}_\pi \left[-\log(\pi(a|s))\right]$ is the causal entropy of policy $\pi$, $\alpha$ is a scale parameter, $\pi_E$ is the policy distribution of the expert. Maximum causal entropy estimation finds the most uniformly distributed policy satisfying feature matching constraints. The feature expectation of the expert policy is used as a statistic to represent the behavior of an expert and is approximated from expert's demonstrations $\mathcal{D} = \{\zeta_0, \cdots, \zeta_N\}$, where $N$ is the number of demonstrations and $\zeta_i$ is a sequence of state and action pairs whose length is $T$, i.e., $\zeta_i = \{(s_0, a_0), \cdots, (s_T, a_T)\}$. In [22], it is shown that the optimal solution of (1) is a softmax distribution.

**Generative Adversarial Imitation Learning**  In [9], Ho and Ermon have extended (1) to a unified framework for IRL by adding a reward regularization as follows:

$$
\max_c \min_{\pi \in \Pi} \quad -\alpha H(\pi) + \mathbb{E}_\pi \left[c(s, a)\right] - \mathbb{E}_{\pi_E} \left[c(s, a)\right] - \psi(c),
\tag{2}
$$

where $c$ is a cost function and $\psi$ is a convex regularization for cost $c$. As shown in [9], many existing IRL methods can be interpreted with this framework, such as MaxEnt IRL [1], apprenticeship learning [11], and multiplicative weights apprenticeship learning [23]. Existing IRL methods based on (2) often require to solve the inner minimization over $\pi$ for fixed $c$ in order to compute the gradient of $c$. In [22], Ziebart showed that the inner minimization is equivalent to a soft Markov decision process (soft MDP) under the reward $-c$ and proposed soft value iteration to solve the soft MDP. However, solving a soft MDP every iteration is often intractable for problems with large state and action spaces and also requires the transition probability which is not accessible in many cases. To address this issue, the generative adversarial imitation learning (GAIL) framework is proposed in [9] to avoid

solving the soft MDP problem directly. The unified imitation learning problem (2) can be converted into the GAIL framework as follows:

$$\min_{\pi \in \Pi} \max_{\mathbf{D}} \quad \mathbb{E}_{\pi} \left[ \log(\mathbf{D}(s,a)) \right] + \mathbb{E}_{\pi_E} \left[ \log(1 - \mathbf{D}(s,a)) \right] - \alpha H(\pi), \tag{3}$$

where $\mathbf{D} \in (0,1)^{|\mathcal{S}||\mathcal{A}|}$ indicates a discriminator, which returns the probability that a given demonstration is from a learner, i.e., 1 for learner's demonstrations and 0 for expert's demonstrations. Notice that we can interpret $\log(\mathbf{D})$ as cost $c$ (or reward of $-c$).

Since existing IRL methods, including GAIL, are often based on the maximum causal entropy, they model the expert's policy using a softmax distribution, which can assign non-zero probability to non-expert actions in a discrete action space. Furthermore, in a continuous action space, expert's behavior is often modeled using a uni-modal Gaussian distribution, which is not proper to model multi-modal behaviors. To handle these issues, we propose a sparsemax distribution as the policy of an expert and provide a natural extension to handle a continuous action space using a mixture density network with sparsemax weight selection.

**Sparse Markov Decision Processes** In [3], a sparse Markov decision process (sparse MDP) is proposed by utilizing the causal sparse Tsallis entropy $W(\pi) \triangleq \frac{1}{2} \mathbb{E}_{\pi} \left[ 1 - \pi(a|s) \right]$ to the expected discounted rewards sum, i.e., $\mathbb{E}_{\pi} \left[ \mathbf{r}(s,a) \right] + \alpha W(\pi)$. Note that $W(\pi)$ is an extension of a special case of the generalized Tsallis entropy, i.e., $S_{k,q}(p) = \frac{k}{q-1} \left( 1 - \sum_i p_i^q \right)$, for $k = \frac{1}{2}, q = 2$, to sequential random variables [1]. It is shown that the optimal policy of a sparse MDP is a sparse and multi-modal policy distribution [3]. Furthermore, sparse Bellman optimality conditions were derived as follows:

$$Q(s,a) \triangleq r(s,a) + \gamma \sum_{s'} V(s') T(s'|s,a), \ \pi(a|s) = \max \left( \frac{Q(s,a)}{\alpha} - \tau \left( \frac{Q(s,\cdot)}{\alpha} \right), 0 \right),$$

$$V(s) = \alpha \left[ \frac{1}{2} \sum_{a \in S(s)} \left( \left( \frac{Q(s,a)}{\alpha} \right)^2 - \tau \left( \frac{Q(s,\cdot)}{\alpha} \right)^2 \right) + \frac{1}{2} \right], \tag{4}$$

where $\tau \left( \frac{Q(s,\cdot)}{\alpha} \right) = \frac{\sum_{a \in S(s)} \frac{Q(s,a)}{\alpha} - 1}{K_s}$, $S(s)$ is a set of actions satisfying $1 + i \frac{Q(s,a_{(i)})}{\alpha} > \sum_{j=1}^{i} \frac{Q(s,a_{(j)})}{\alpha}$ with $a_{(i)}$ indicating the action with the $i$th largest state-action value $Q(s,a)$, and $K_s$ is the cardinality of $S(s)$. In [3], a sparsemax policy shows better performance compared to a softmax policy since it assigns zero probability to non-optimal actions whose state-action value is below the threshold $\tau$. In this paper, we utilize this property in imitation learning by modeling expert's behavior using a sparsemax distribution. In Section 4, we show that the optimal solution of an MCTE problem also has a sparsemax distribution and, hence, the optimality condition of sparse MDPs is closely related to that of MCTE problems.

## 4   Principle of Maximum Causal Tsallis Entropy

In this section, we formulate maximum causal Tsallis entropy imitation learning (MCTEIL) and show that MCTE induces a sparse and multi-modal distribution which has an adaptable supporting set. The problem of maximizing the causal Tsallis entropy $W(\pi)$ can be formulated as follows:

$$\begin{aligned} \underset{\pi \in \Pi}{\text{maximize}} \quad & \alpha W(\pi) \\ \text{subject to} \quad & \mathbb{E}_{\pi} \left[ \phi(s,a) \right] = \mathbb{E}_{\pi_E} \left[ \phi(s,a) \right]. \end{aligned} \tag{5}$$

In order to derive optimality conditions, we will first change the optimization variable from a policy distribution to a state-action visitation measure. Then, we prove that the MCTE problem is concave with respect to the visitation measure. The necessary and sufficient conditions for an optimal solution are derived from the Karush-Kuhn-Tucker (KKT) conditions using the strong duality and the optimal policy is shown to be a sparsemax distribution. Furthermore, we also provide an interesting interpretation of the MCTE framework as robust Bayes estimation in terms of the Brier score. Hence,

the proposed method can be viewed as maximization of the worst case performance in the sense of the Brier score [24].

We can change the optimization variable from a policy distribution to a state-action visitation measure based on the following theorem.

**Theorem 1 (Theorem 2 of Syed et al. [25])** *Let* $\mathbf{M}$ *be a set of state-action visitation measures, i.e.,* $\mathbf{M} \triangleq \{\rho | \forall s, \ a, \ \rho(s,a) \geq 0, \ \sum_a \rho(s,a) = d(s) + \gamma \sum_{s',a'} T(s|s',a')\rho(s',a')\}$. *If* $\rho \in \mathbf{M}$, *then it is a state-action visitation measure for* $\pi_\rho(a|s) \triangleq \frac{\rho(s,a)}{\sum_a \rho(s,a)}$, *and* $\pi_\rho$ *is the unique policy whose state-action visitation measure is* $\rho$.

The proof of Theorem 1 can be found in [25] or in Puterman [26]. Theorem 1 guarantees the one-to-one correspondence between a policy distribution and state-action visitation measure. Then, the objective function $W(\pi)$ is converted into the function of $\rho$ as follows.

**Theorem 2** *Let* $\bar{W}(\rho) = \frac{1}{2} \sum_{s,a} \rho(s,a) \left(1 - \frac{\rho(s,a)}{\sum_{a'} \rho(s,a')}\right)$. *Then, for any stationary policy* $\pi \in \Pi$ *and any state-action visitation measure* $\rho \in \mathbf{M}$, $W(\pi) = \bar{W}(\rho_\pi)$ *and* $\bar{W}(\rho) = W(\pi_\rho)$ *hold.*

The proof is provided in the supplementary material. Theorem 2 tells us that if $\bar{W}(\rho)$ has the maximum at $\rho^*$, then $W(\pi)$ also has the maximum at $\pi_{\rho^*}$. Based on Theorem 1 and 2, we can freely convert the problem (5) into

$$\begin{aligned} \underset{\rho \in \mathbf{M}}{\text{maximize}} \quad & \alpha \bar{W}(\rho) \\ \text{subject to} \quad & \sum_{s,a} \rho(s,a)\phi(s,a) = \sum_{s,a} \rho_E(s,a)\phi(s,a), \end{aligned} \tag{6}$$

where $\rho_E$ is the state-action visitation measure corresponding to $\pi_E$.

### 4.1 Optimality Condition of Maximum Causal Tsallis Entropy

We show that the optimal policy of the problem (6) is a sparsemax distribution using the KKT conditions. In order to use the KKT conditions, we first show that the MCTE problem is concave.

**Theorem 3** $\bar{W}(\rho)$ *is strictly concave with respect to* $\rho \in \mathbf{M}$.

The proof of Theorem 3 is provided in the supplementary material. Since all constraints are linear and the objective function is concave, (6) is a concave problem and, hence, strong duality holds. The dual problem is defined as follows:

$$\begin{aligned} \max_{\theta,c,\lambda} \ \min_{\rho} \quad & L_W(\theta,c,\lambda,\rho) \\ \text{subject to} \quad & \forall s,a \ \ \lambda_{sa} \geq 0, \end{aligned} \tag{7}$$

where $L_W(\theta,c,\lambda,\rho) = -\alpha\bar{W}(\rho) - \sum_{s,a} \rho(s,a)\theta^\intercal \phi(s,a) + \sum_{s,a} \rho_E(s,a)\theta^\intercal \phi(s,a) - \sum_{s,a} \lambda_{sa}\rho(s,a) + \sum_s c_s \left(\sum_a \rho(s,a) - d(s) - \gamma \sum_{s',a'} T(s|s',a')\rho(s',a')\right)$ and $\theta$, $c$, and $\lambda$ are Lagrangian multipliers and the constraints come from $\mathbf{M}$. Then, the optimal solution of primal and dual variables necessarily and sufficiently satisfy the KKT conditions.

**Theorem 4** *The optimal solution of (6) sufficiently and necessarily satisfies the following conditions:*

$$q_{sa} \triangleq \theta^\intercal \phi(s,a) + \gamma \sum_{s'} c_{s'} T(s'|s,a), \ c_s = \alpha \left[\frac{1}{2} \sum_{a \in S(s)} \left(\left(\frac{q_{sa}}{\alpha}\right)^2 - \tau\left(\frac{q_s}{\alpha}\right)^2\right) + \frac{1}{2}\right],$$

$$and \quad \pi_\rho(a|s) = \max\left(\frac{q_{sa}}{\alpha} - \tau\left(\frac{q_s}{\alpha}\right), 0\right),$$

*where* $\pi_\rho(a|s) = \frac{\rho(s,a)}{\sum_a \rho(s,a)}$, $q_{sa}$ *is an auxiliary variable, and* $q_s = [q_{sa_1} \cdots q_{sa_{|\mathcal{A}|}}]^\intercal$.

The optimality conditions of the problem (6) tell us that the optimal policy is a sparsemax distribution which assigns zero probability to an action whose auxiliary variable $q_{sa}$ is below the threshold $\tau$,

---

**Algorithm 1** Maximum Causal Tsallis Entropy Imitation Learning

---
1: Expert's demonstrations $\mathcal{D}$ are given
2: Initialize policy and discriminator parameters $\nu, \omega$
3: **while** until convergence **do**
4:     Sample trajectories $\{\zeta\}$ from $\pi_\nu$
5:     Update $\omega$ with the gradient of $\sum_{\{\zeta\}} \log(\mathbf{D}_\omega(s,a)) + \sum_{\mathcal{D}} \log(1 - \mathbf{D}_\omega(s,a))$.
6:     Update $\nu$ using a policy optimization method with reward function $-\mathbb{E}_{\pi_\nu}\left[\log(\mathbf{D}_\omega(s,a))\right] + \alpha W(\pi_\nu)$
7: **end while**

---

which determines a supporting set. If expert's policy is multi-modal at state $s$, the resulting $\pi_\rho(\cdot|s)$ becomes multi-modal and induces a multi-modal distribution with a large supporting set. Otherwise, the resulting policy has a sparse and smaller supporting set. Therefore, a sparsemax policy has advantages over a softmax policy for modeling sparse and multi-modal behaviors of an expert whose supporting set varies according to the state.

Furthermore, we also discover an interesting connection between the optimality condition of an MCTE problem and the sparse Bellman optimality condition (4). Since the optimality condition is equivalent to the sparse Bellman optimality equation [3], we can compute the optimal policy and Lagrangian multiplier $c_s$ by solving a sparse MDP under the reward function $\mathbf{r}(s,a) = \theta^{*\mathsf{T}}\phi(s,a)$, where $\theta^*$ is the optimal dual variable. In addition, $c_s$ and $q_{sa}$ can be viewed as a state value and state-action value for the reward $\theta^{*\mathsf{T}}\phi(s,a)$, respectively.

### 4.2 Interpretation as Robust Bayes

In this section, we provide an interesting interpretation about the MCTE framework. In general, maximum entropy estimation can be viewed as a minimax game between two players. One player is called a decision maker and the other player is called the nature, where the nature assigns a distribution to maximize the decision maker's misprediction while the decision maker tries to minimize it [27]. The same interpretation can be applied to the MCTE framework. We show that the proposed MCTE problem is equivalent to a minimax game with the Brier score [24].

**Theorem 5** *The maximum causal Tsallis entropy distribution minimizes the worst case prediction Brier score,*

$$\min_{\pi \in \Pi} \max_{\tilde{\pi} \in \Pi} \mathbb{E}_{\tilde{\pi}}\left[\sum_{a'} \frac{1}{2}\left(\mathbb{1}_{\{a'=a\}} - \pi(a|s)\right)^2\right] \quad \text{subject to} \quad \mathbb{E}_\pi[\phi(s,a)] = \mathbb{E}_{\pi_E}[\phi(s,a)] \tag{8}$$

*where $\sum_{a'} \frac{1}{2}\left(\mathbb{1}_{\{a'=a\}} - \pi(a|s)\right)^2$ is the Brier score.*

Note that minimizing the Brier score minimizes the misprediction ratio while we call it a score here. Theorem 5 is a straightforward extension of the robust Bayes results in [27] to sequential decision problems. This theorem tells us that the MCTE problem can be viewed as a minimax game between a sequential decision maker $\pi$ and the nature $\tilde{\pi}$ based on the Brier score. In this regards, the resulting estimator can be interpreted as the best decision maker against the worst that the nature can offer.

## 5 Maximum Causal Tsallis Entropy Imitation Learning

In this section, we propose a maximum causal Tsallis entropy imitation learning (MCTEIL) algorithm to solve a model-free IL problem in a continuous action space. In many real-world problems, state and action spaces are often continuous and transition probability of a world cannot be accessed. To apply the MCTE framework for a continuous space and model-free case, we follow the extension of GAIL [9], which trains a policy and reward alternatively, instead of solving RL at every iteration. We extend the MCTE framework to a more general case with reward regularization and it is formulated by replacing the causal entropy $H(\pi)$ in the problem (2) with the causal Tsallis entropy $W(\pi)$ as follows:

$$\max_\theta \min_{\pi \in \Pi} \quad -\alpha W(\pi) - \mathbb{E}_\pi[\theta^\mathsf{T}\phi(s,a)] + \mathbb{E}_{\pi_E}[\theta^\mathsf{T}\phi(s,a)] - \psi(\theta). \tag{9}$$

Similarly to [9], we convert the problem (9) into the generative adversarial setting as follows.

**Theorem 6** *The maximum causal sparse Tsallis entropy problem (9) is equivalent to the problem:*

$$\min_{\pi \in \Pi} \ \psi^* \left( \mathbb{E}_\pi \left[ \phi(s,a) \right] - \mathbb{E}_{\pi_E} \left[ \phi(s,a) \right] \right) - \alpha W(\pi),$$

*where $\psi^*(x) = \sup_y \{ y^\intercal x - \psi(y) \}$.*

The proof is detailed in the supplementary material. The proof of Theorem 6 depends on the fact that the objective function of (9) is concave with respect to $\rho$ and is convex with respect to $\theta$. Hence, we first switch the optimization variables from $\pi$ to $\rho$ and, using the minimax theorem [28], the maximization and minimization are interchangeable and the generative adversarial setting is derived. Similarly to [9], Theorem 6 says that a MCTE problem can be interpreted as minimization of the distance between expert's feature expectation and training policy's feature expectation, where $\psi^*(x_1 - x_2)$ is a proper distance function since $\psi(x)$ is a convex function. Let $e_{sa} \in \mathbb{R}^{|\mathcal{S}||\mathcal{A}|}$ be a feature indicator vector, such that the $sa$th element is one and zero elsewhere. If we set $\psi$ to $\psi_{GA}(\theta) \triangleq \mathbb{E}_{\pi_E}[g(\theta^\intercal e_{sa})]$, where $g(x) = -x - \log(1 - e^x)$ for $x < 0$ and $g(x) = \infty$ for $x \geq 0$, we can convert the MCTE problem into the following generative adversarial setting:

$$\min_{\pi \in \Pi} \max_{\mathbf{D}} \quad \mathbb{E}_\pi \left[ \log(\mathbf{D}(s,a)) \right] + \mathbb{E}_{\pi_E} \left[ \log(1 - \mathbf{D}(s,a)) \right] - \alpha W(\pi), \tag{10}$$

where $\mathbf{D}$ is a discriminator. The problem (10) can be solved by MCTEIL which consists of three steps. First, trajectories are sampled from the training policy $\pi_\nu$ and discriminator $\mathbf{D}_\omega$ is updated to distinguish whether the trajectories are generated by $\pi_\nu$ or $\pi_E$. Finally, the training policy $\pi_\nu$ is updated with a policy optimization method under the sum of rewards $\mathbb{E}_\pi \left[ -\log(\mathbf{D}_\omega(s,a)) \right]$ with a causal Tsallis entropy bonus $\alpha W(\pi_\nu)$. The algorithm is summarized in Algorithm 1.

**Sparse Mixture Density Network**    We further employ a novel mixture density network (MDN) with sparsemax weight selection, which can model sparse and multi-modal behavior of an expert, which is called a sparse MDN. In many imitation learning algorithms, a Gaussian network is often employed to model expert's policy in a continuous action space. However, a Gaussian distribution is inappropriate to model the multi-modality of an expert since it has a single mode. An MDN is more suitable for modeling a multi-modal distribution. In particular, a sparse MDN is a proper extension of a sparsemax distribution for a continuous action space. The input of a sparse MDN is state $s$ and the output of a sparse MDN is components of $K$ mixtures of Gaussians: mixture weights $\{w_i\}$, means $\{\mu_i\}$, and covariance matrices $\{\Sigma_i\}$. A sparse MDN policy is defined as

$$\pi(a|s) = \sum_i^K w_i(s) \mathcal{N}(a; \mu_i(s), \Sigma_i(s)),$$

where $\mathcal{N}(a; \mu, \Sigma)$ indicates a multivariate Gaussian density at point $a$ with mean $\mu$ and covariance $\Sigma$. In our implementation, $w(s)$ is computed as a sparsemax distribution, while most existing MDN implementations utilize a softmax distribution. Modeling the expert's policy using an MDN with $K$ mixtures can be interpreted as separating continuous action space into $K$ representative actions. Since we model mixture weights using a sparsemax distribution, the number of mixtures used to model the expert's policy can vary depending on the state. In this regards, the sparsemax weight selection has an advantage over the soft weight selection since the former utilizes mixture components more efficiently as unnecessary components will be assigned with zero weights.

**Tsallis Entropy of Mixture Density Network**    An interesting fact is that the causal Tsallis entropy of an MDN has an analytic form while the Gibbs-Shannon entropy of an MDN is intractable.

**Theorem 7** *Let $\pi(a|s) = \sum_i^K w_i(s)\mathcal{N}(a; \mu_i(s), \Sigma_i(s))$ and $\rho_\pi(s) = \sum_a \rho_\pi(s,a)$. Then,*

$$W(\pi) = \frac{1}{2} \sum_s \rho_\pi(s) \left( 1 - \sum_i^K \sum_j^K w_i(s)w_j(s)\mathcal{N}\left( \mu_i(s); \mu_j(s), \Sigma_i(s) + \Sigma_j(s) \right) \right). \tag{11}$$

The proof is included in the supplementary material. The analytic form of the Tsallis entropy shows that the Tsallis entropy is proportional to the distance between mixture means. Hence, maximizing the Tsallis entropy of a sparse MDN encourages exploration of diverse directions during the policy optimization step of MCTEIL. In imitation learning, the main benefit of the generative adversarial

setting is that the resulting policy is more robust than that of supervised learning since it can learn how to recover from a less demonstrated region to a demonstrated region by exploring the state-action space during training. Maximum Tsallis entropy of a sparse MDN encourages efficient exploration by giving bonus rewards when mixture means are spread out. (11) also has an effect of utilizing mixtures more efficiently by penalizing for modeling a single mode using several mixtures. Consequently, the Tsallis entropy $W(\pi)$ has clear benefits in terms of both exploration and mixture utilization.

## 6 Experiments

To verify the effectiveness of the proposed method, we compare MCTEIL with several other imitation learning methods. First, we use behavior cloning (BC) as a baseline. Second, generative adversarial imitation learning (GAIL) with a single Gaussian distribution is compared. We also compare a straightforward extension of GAIL for a multi-modal policy by using a softmax weighted mixture density network (soft MDN) in order to validate the efficiency of the proposed sparse-max weighted MDN. In soft GAIL, due to the intractability of the causal entropy of a mixture of Gaussians, we approximate the entropy term by adding $-\alpha \log(\pi(a_t|s_t))$ to $-\log(\mathbf{D}(s_t, a_t))$ since $\mathbb{E}_\pi[-\log(\mathbf{D}(s,a))] + \alpha H(\pi) = \mathbb{E}_\pi[-\log(\mathbf{D}(s,a)) - \alpha \log(\pi(a|s))]$. We also compare info GAIL [21] which learns simultaneously both policy and the latent structure of experts' demonstrations. In info GAIL, a posterior distribution of a latent code is learned to cluster multi-modal demonstrations. The posterior distribution is trained to consistently assign the latent code to similar demonstrations and Once the latent codes are assigned to the demonstrations, the policy function conditioned on a latent code is trained to generate the corresponding demonstrations. Different modes in demonstrations are captured by assigning different latent codes.

### 6.1 Multi-Goal Environment

To validate that the proposed method can learn multi-modal behavior of an expert, we design a simple multi-goal environment with four attractors and four repulsors, where an agent tries to reach one of attractors while avoiding all repulsors as shown in Figure 1(a). The agent follows the point-mass dynamics and get a positive reward (resp., a negative reward) when getting closer to an attractor (resp., repulsor). Intuitively, this problem has multi-modal optimal actions at the center. We first train the optimal policy using [3] and generate 300 demonstrations from the expert's policy. For tested methods, 500 episodes are sampled at each iteration. In every iteration, we measure the average return using the underlying rewards and the reachability which is measured by counting how many goals are reached. If the algorithm captures the multi-modality of expert's demonstrations, then, the resulting policy will show high reachability. All algorithms run repeatedly with seven different random seeds.

The results are shown in Figure 1(b) and 1(c). Since the rewards are multi-modal, it is easy to get a high return if the algorithm learns only uni-modal behavior. Hence, the average returns of soft GAIL, info GAIL and MCTEIL increases similarly. However, when it comes to the reachability, MCTEIL outperforms other methods when they use the same number of mixtures. In particular, MCTEIL can learn all modes in demonstrations at the end of learning while soft GAIL and info GAIL suffer from collapsing modes. This advantage clearly comes from the maximum Tsallis entropy of a sparse MDN since the analytic form of the Tsallis entropy directly penalizes collapsed mixture means while $-\log(\pi(a|s))$ indirectly prevents modes collapsing in soft GAIL. Furthermore, info-GAIL also shows mode collapsing while the proposed method can learn every modes. Since info-GAIL has to train a posterior distribution over the latent code to separate demonstrations, it requires more iterations for reaching all modes as well as prone to the mode collapsing problems. Consequently, we can conclude that the MCTEIL efficiently utilizes each mixture for wide-spread exploration.

### 6.2 Continuous Control Environment

We test MCTEIL with a sparse MDN on MuJoCo [10], which is a physics-based simulator, using *Halfcheetah, Walker2d, Reacher*, and *Ant*. We train the expert policy distribution using trust region policy optimization (TRPO) [29] under the true reward function and generate 50 demonstrations from the expert policy. We run algorithms with varying numbers of demonstrations, $4, 11, 18$, and $25$, and all experiments have been repeated three times with different random seeds. To evaluate the performance of each algorithm, we sample $50$ episodes from the trained policy and measure the

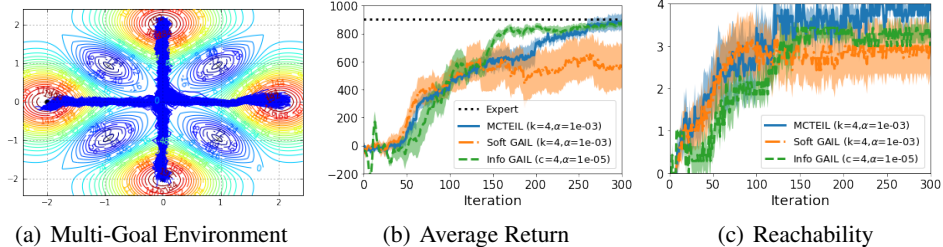

| (a) Multi-Goal Environment | (b) Average Return | (c) Reachability |

Figure 1: (a) The environment and multi-modal demonstrations are shown. The contour shows the underlying reward map. (b) The average return during training. (c) The reachability during training, where $k$ is the number of mixtures, $c$ is a dimension of the latent code, and $\alpha$ is a regularization coefficient.

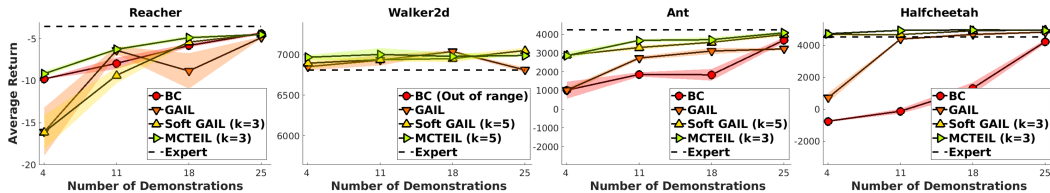

Figure 2: Average returns of trained policies. For soft GAIL and MCTEIL, $k$ indicates the number of mixture and $\alpha$ is an entropy regularization coefficient. A dashed line indicates the performance of an expert.

average return value using the underlying rewards. For methods using an MDN, we use the best number of mixtures using a brute force search.

The results are shown in Figure 2. For three problems, except Walker2d, MCTEIL outperforms the other methods with respect to the average return as the number of demonstrations increases. For Walker2d, MCTEIL and soft GAIL show similar performance. Especially, in the reacher problem, we obtain the similar results reported in [9], where BC works better than GAIL. However, our method shows the best performance for all demonstration counts. It is observed that the MDN policy tends to show high performance consistently since MCTEIL and soft GAIL are consistently ranked within the top two high performing algorithms. From these results, we can conclude that an MDN policy explores better than a single Gaussian policy since an MDN can keep searching multiple directions during training. In particular, since the maximum Tsallis entropy makes each mixture mean explore in different directions and a sparsemax distribution assigns zero weight to unnecessary mixture components, MCTEIL efficiently explores and shows better performance compared to soft GAIL with a soft MDN. Consequently, we can conclude that MCTEIL outperforms other imitation learning methods and the causal Tsallis entropy has benefits over the causal Gibbs-Shannon entropy as it encourages exploration more efficiently.

## 7 Conclusion

In this paper, we have proposed a novel maximum causal Tsallis entropy (MCTE) framework, which induces a sparsemax distribution as the optimal solution. We have also provided the full mathematical analysis of the proposed framework, including the concavity of the problem, the optimality condition, and the interpretation as robust Bayes. We have also developed the maximum causal Tsallis entropy imitation learning (MCTEIL) algorithm, which can efficiently solve a MCTE problem in a continuous action space since the Tsallis entropy of a mixture of Gaussians encourages exploration and efficient mixture utilization. In experiments, we have verified that the proposed method has advantages over existing methods for learning the multi-modal behavior of an expert since a sparse MDN can search in diverse directions efficiently. Furthermore, the proposed method has outperformed BC, GAIL, and GAIL with a soft MDN on the standard IL problems in the MuJoCo environment. From the analysis and experiments, we have shown that the proposed MCTEIL method is an efficient and principled way to learn the multi-modal behavior of an expert.

**Acknowledgments**

This work was supported in part by Basic Science Research Program through the National Research Foundation of Korea (NRF) funded by the Ministry of Science and ICT (NRF-2017R1A2B2006136) and by the Brain Korea 21 Plus Project in 2018.

## Footnotes

[1] The casual entropy is generally defined upon causally conditioned random variables. However, in this paper, the causal Tsallis entropy is defined over the random variables with Markov properties, i.e., $\pi(a_t|s_t) = \pi(a_t|s_t, a_{t-1}, s_{t-1}, \cdots, a_0, s_0)$, since we only consider an MDP.

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
