[Supplementary Material]

# Maximum Tsallis Entropy Imitation Learning: Supplementary Material

**Kyungjae Lee**[1], **Sungjoon Choi**[2], **and Songhwai Oh**[1]

Dep. of Electrical and Computer Engineering, Seoul National University[1]

Kakao Brain[2]

kyungjae.lee@rllab.snu.ac.kr, sam.choi@kakaobrain.com,
songhwai@snu.ac.kr

In this supplementary material, we provide proofs of theorems in the main paper and implementation details of our algorithms. We first show that the maximum causal Tsallis entropy (MCTE) problem is concave with respect to the state-action visitation $\rho(s, a)$. Then, the optimality condition of an MCTE problem is derived from Karush–Kuhn–Tucker (KKT) conditions. We also show that the proposed framework can be interpreted as robust Bayes estimation with the Brier score. Finally, to convert our problem into the generative adversarial training framework, the interchangability of maximization and minimization is proved.

## 1 Analysis

We consider the maximum causal Tsallis entropy problem defined as follows:

$$
\begin{aligned}
\underset{\pi}{\text{maximize}} \quad & \alpha W(\pi) \\
\text{subject to} \quad & \mathbb{E}_\pi \left[ \phi(s, a) \right] = \mathbb{E}_{\pi_E} \left[ \phi(s, a) \right], \\
& \forall s, a \quad \sum_{a'} \pi(a'|s) = 1, \quad \pi(a|s) \geq 0.
\end{aligned}
\tag{1}
$$

Note that the constraints for $\Pi$ are explicitly added. For the remainder of this supplementary material, we will explicitly write all constraints for $\Pi$ and $\mathbf{M}$.

### 1.1 Change of Variables

Before proving the concavity of (1), we first see two important theorem as follows:

**Theorem 1** (Theorem 2 of Syed et al. [1]). *If $\rho$ satisfies Bellman flow constraints, then it is a state action visitation for $\pi_\rho(a|s) \triangleq \frac{\rho(s,a)}{\sum_a \rho(s,a)}$, and $\pi_\rho$ is the unique policy whose state action visitation is $\rho$ where Bellman flow constraints are defined as $\sum_a \rho(s, a) = d(s) + \sum_{s',a'} T(s|s', a')\rho(s', a')$.*

*Proof.* The proof can be found in [1] or in Puterman [2]. □

**Theorem 2.** *Let $\bar{W}(\rho) = \frac{1}{2} \sum_{s,a} \rho(s, a) \left( 1 - \frac{\rho(s,a)}{\sum_{a'} \rho(s,a')} \right)$. Then, for any stationary policy $\pi \in \Pi$ and any state-action visitation measure $\rho \in \mathbf{M}$, $W(\pi) = \bar{W}(\rho_\pi)$ and $\bar{W}(\rho) = W(\pi_\rho)$ hold.*

*Proof.* The proof is simply done by checking two equalities. First,

$$
\begin{aligned}
W(\pi) &= \frac{1}{2} \mathbb{E}_\pi \left[ 1 - \pi(a|s) \right] = \frac{1}{2} \sum_{s,a} \rho_\pi(s, a) \left( 1 - \pi(a|s) \right) \\
&= \frac{1}{2} \sum_{s,a} \rho_\pi(s, a) \left( 1 - \frac{\rho_\pi(s, a)}{\sum_{a'} \rho_\pi(s, a')} \right)
\end{aligned}
$$

and, second,

$$\bar{W}(\rho) = \frac{1}{2} \sum_{s,a} \rho(s,a) \left(1 - \frac{\rho(s,a)}{\sum_{a'} \rho(s,a')}\right) = \frac{1}{2} \sum_{s,a} \rho_{\pi_\rho}(s,a) \left(1 - \pi_\rho(a|s)\right)$$
$$= W(\pi_\rho).$$

$\square$

Base on Theorem 1 and Theorem 2, we convert the problem (1) into

$$
\begin{aligned}
\underset{\rho}{\text{maximize}} \quad & \alpha \bar{W}(\rho) \\
\text{subject to} \quad & \sum_{s,a} \rho(s,a)\phi(s,a) = \sum_{s,a} \rho_E(s,a)\phi(s,a), \\
& \forall\, s,a, \ \ \rho(s,a) \geq 0, \ \sum_a \rho(s,a) = d(s) + \gamma \sum_{s',a'} T(s|s',a')\rho(s',a')
\end{aligned}
\tag{2}
$$

where $\bar{W}(\rho) = W(\frac{\rho}{\sum_a \rho})$, the second constraints are Bellman flow constraints for $\mathbf{M}$, and $\rho_E$ is the state action visitation corresponding to $\pi_E$.

## 1.2 Concavity of Maximum causal Tsallis Entropy

The following theorem shows that the objective function $\bar{W}(\rho)$ of the problem (2) is a concave function.

**Theorem 3.** $\bar{W}(\rho)$ *is strictly concave with respect to* $\rho \in \mathbf{M}$.

*Proof.* Proof of concavity of $\bar{W}(\rho)$ is equivalent to show that following inequality is satisfied for all state $s$ and action $a$ pairs:

$$
(\lambda_1 \rho_1(s,a) + \lambda_2 \rho_2(s,a)) \left(1 - \frac{\lambda_1 \rho_1(s,a) + \lambda_2 \rho_2(s,a)}{\lambda_1 \sum_{a'} \rho_1(s,a') + \lambda_2 \sum_{a'} \rho_2(s,a')}\right)
$$
$$
\geq \lambda_1 \rho_1(s,a) \left(1 - \frac{\rho_1(s,a)}{\sum_{a'} \rho_1(s,a')}\right) + \lambda_2 \rho_2(s,a) \left(1 - \frac{\rho_2(s,a)}{\sum_{a'} \rho_2(s,a')}\right)
$$

where $\lambda_1 \geq 0$, $\lambda_2 \geq 0$, and $\lambda_1 + \lambda_2 = 1$. For notational simplicity, $\rho_i(s,a)$ and $\sum_{a'} \rho_i(s,a')$ are replaced with $a_i$ and $b_i$, respectively. Then, the right-hand side is

$$
\sum_{i=1,2} \lambda_i a_i \left(1 - \frac{a_i}{b_i}\right) = \sum_{i=1,2} \lambda_i a_i \left(1 - \frac{\lambda_i a_i}{\lambda_i b_i}\right)
$$
$$
= \left(\sum_{j=1,2} \lambda_j b_j\right) \sum_{i=1,2} \left[\frac{\lambda_i b_i}{\left(\sum_{j=1,2} \lambda_j b_j\right)} \frac{\lambda_i a_i}{\lambda_i b_i} \left(1 - \frac{\lambda_i a_i}{\lambda_i b_i}\right)\right].
$$

Let $F(x) = x(1-x)$, which is a concave function. Then the above equation can be expressed as follows,

$$
\sum_{i=1,2} \lambda_i a_i \left(1 - \frac{a_i}{b_i}\right) = \left(\sum_{j=1,2} \lambda_j b_j\right) \sum_{i=1,2} \left[\frac{\lambda_i b_i}{\left(\sum_{j=1,2} \lambda_j b_j\right)} F\left(\frac{\lambda_i a_i}{\lambda_i b_i}\right)\right].
$$

By using the property of concave function $F(x)$[1], we obtain the following inequality:

$$\left(\sum_{j=1,2}\lambda_j b_j\right)\sum_{i=1,2}\left[\frac{\lambda_i b_i}{\left(\sum_{j=1,2}\lambda_j b_j\right)}F\left(\frac{\lambda_i a_i}{\lambda_i b_i}\right)\right]$$

$$\leq\left(\sum_{j=1,2}\lambda_j b_j\right)F\left(\sum_{i=1,2}\left[\frac{\lambda_i b_i}{\left(\sum_{j=1,2}\lambda_j b_j\right)}\frac{\lambda_i a_i}{\lambda_i b_i}\right]\right)=\left(\sum_{j=1,2}\lambda_j b_j\right)F\left(\frac{\sum_{i=1,2}\lambda_i a_i}{\sum_{j=1,2}\lambda_j b_j}\right)$$

$$=\left(\sum_{j=1,2}\lambda_j b_j\right)\frac{\sum_{i=1,2}\lambda_i a_i}{\sum_{j=1,2}\lambda_j b_j}\left(1-\frac{\sum_{i=1,2}\lambda_i a_i}{\sum_{j=1,2}\lambda_j b_j}\right)=\sum_{i=1,2}\lambda_i a_i\left(1-\frac{\sum_{i=1,2}\lambda_i a_i}{\sum_{j=1,2}\lambda_j b_j}\right).$$

Finally, we have the following inequality for every state and action pair,

$$(\lambda_1\rho_1(s,a)+\lambda_2\rho_2(s,a))\left(1-\frac{\lambda_1\rho_1(s,a)+\lambda_2\rho_2(s,a)}{\lambda_1\sum_{a'}\rho_1(s,a')+\lambda_2\sum_{a'}\rho_2(s,a')}\right)$$

$$\geq\lambda_1\rho_1(s,a)\left(1-\frac{\rho_1(s,a)}{\sum_{a'}\rho_1(s,a')}\right)+\lambda_2\rho_2(s,a)\left(1-\frac{\rho_2(s,a)}{\sum_{a'}\rho_2(s,a')}\right),$$

and, by summing up with respect to $s,a$, we get

$$\bar{W}(\lambda_1\rho_1+\lambda_2\rho_2)\geq\lambda_1\bar{W}(\rho_1)+\lambda_2\bar{W}(\rho_2).$$

Therefore, $\bar{W}(\rho)$ is a concave function. □

Theorem 3 tells us that the problem (2) is a concave problem and, hence, strong duality holds. The dual problem can be found as follows:

$$\begin{aligned}\max_{\theta,c,\lambda}\min_{\rho}\quad & L_W(\theta,c,\lambda,\rho)\\ \text{subject to}\quad & \forall\, s,a,\ \ \lambda(s,a)\geq 0\end{aligned}\tag{3}$$

where $L_W(\theta,c,\lambda,\rho)=-\alpha\bar{W}(\rho)-\sum_{s,a}\rho(s,a)\theta^\mathsf{T}\phi(s,a)+\sum_{s,a}\rho_E(s,a)\theta^\mathsf{T}\phi(s,a)-\sum_{s,a}\lambda_{sa}\rho(s,a)+\sum_s c_s\left(\sum_a\rho(s,a)-d(s)-\gamma\sum_{s',a'}T(s|s',a')\rho(s',a')\right)$ and $\theta$, $c$, and $\lambda$ are Lagrangian multipliers. Since strong duality holds, the optimal solutions of primal and dual variables necessarily and sufficiently satisfy the KKT conditions.

## 1.3 Optimality Condition from Karush–Kuhn–Tucker (KKT) conditions

The following theorem explains the optimality condition of the maximum causal Tsallis entropy problem and also tells us that the optimal policy distribution has a sparse and multi-modal distribution.

**Theorem 4.** *The optimal solution of (2) sufficiently and necessarily satisfies the following condition:*

$$q_{sa}\triangleq\theta^\mathsf{T}\phi(s,a)+\gamma\sum_{s'}c_{s'}T(s'|s,a),$$

$$c_s=\alpha\left[\frac{1}{2}\sum_{a\in S(s)}\left(\left(\frac{q_{sa}}{\alpha}\right)^2-\tau\left(\frac{q_s}{\alpha}\right)^2\right)+\frac{1}{2}\right],\textit{and}$$

$$\pi_\rho(a|s)=\max\left(\frac{q_{sa}}{\alpha}-\tau\left(\frac{q_s}{\alpha}\right),0\right),$$

*where $\pi_\rho(a|s)=\frac{\rho(s,a)}{\sum_a\rho(s,a)}$, $q_{sa}$ is an auxiliary variable, and $q_s=[q_{sa_1}\cdots q_{sa_{|\mathcal{A}|}}]^\mathsf{T}$.*

*Proof.* These conditions are derived from the stationary condition of KKT, where the derivative of $L_W$ is equal to zero,

$$\frac{\partial L_W}{\partial\rho(s,a)}=0.$$

We first compute the derivative of $\bar{W}$ as follows:

$$\frac{\partial \bar{W}}{\partial \rho(s,a)} = \frac{1}{2} - \frac{\rho(s,a)}{\sum_{a'} \rho(s,a')} + \frac{1}{2}\sum_{a'}\left(\frac{\rho(s,a')}{\sum_{a'}\rho(s,a')}\right)^2.$$

We also check the derivative of Bellman flow constraints as follows:

$$\frac{\partial \sum_s c_s \left(\sum_{a'}\rho(s,a') - d(s) - \gamma \sum_{s',a'} T(s|s',a')\rho(s',a')\right)}{\partial \rho(s'',a'')} = c_{s''} - \gamma \sum_s c_s T(s|s'',a'').$$

Hence, the stationary condition can be obtained as

$$\begin{aligned}
\frac{\partial L_W}{\partial \rho(s,a)} =& \alpha \left[-\frac{1}{2} + \frac{\rho(s,a)}{\sum_{a'}\rho(s,a')} - \frac{1}{2}\sum_{a'}\left(\frac{\rho(s,a')}{\sum_{a'}\rho(s,a')}\right)^2\right] - \theta^\mathsf{T}\phi(s,a) \\
& + c_s - \gamma \sum_{s'} c_{s'} T(s'|s,a) - \lambda_{sa} = 0.
\end{aligned} \tag{4}$$

First, let us consider a positive $a \in S(s) = \{a|\rho(s,a) > 0\}$. From the complementary slackness, the corresponding $\lambda_{sa}$ is zero. By replacing $\frac{\rho(s,a)}{\sum_a \rho(s,a')}$ with $\pi_\rho(a|s)$ and using the definition of $q_{sa}$, the following equation is obtained from the stationary condition (4).

$$\pi(a|s) - \frac{q_{sa}}{\alpha} = \frac{1}{2} + \frac{1}{2}\sum_{a'}\left(\pi(a'|s)\right)^2 - \frac{c_s}{\alpha}. \tag{5}$$

It can be observed that the right hand side of the equation only depends on the state $s$ and is constant for the action $a$. In this regards, by summing up with respect to the action with positive $\rho(s,a) > 0$, $c_s$ is obtained as follows:

$$1 - \sum_{a\in S(s)}\frac{q_{sa}}{\alpha} = K\left(\frac{1}{2} + \frac{1}{2}\sum_{a'}\left(\pi(a'|s)\right)^2 - \frac{c_s}{\alpha}\right)$$

$$\frac{c_s}{\alpha} = \frac{1}{2} + \frac{1}{2}\sum_{a'}\left(\pi(a'|s)\right)^2 + \frac{\sum_{a\in S(s)}\frac{q_{sa}}{\alpha} - 1}{K},$$

where $K$ is the number of actions with positive $\rho(s,a) > 0$. By plug in $\frac{c_s}{\alpha}$ into (5), we obtain a policy as follows:

$$\pi(a|s) = \frac{q_{sa}}{\alpha} - \left(\frac{\sum_{a\in S(s)}\frac{q_{sa}}{\alpha} - 1}{K}\right)$$

Now, we define $\tau(\frac{q_s}{\alpha}) \triangleq \frac{\sum_{a\in S(s)}\frac{q_{sa}}{\alpha} - 1}{K}$, and, interestingly, $\tau$ is the same as the threshold of a sparsemax distribution [3]. Then, we can obtain the optimality condition for the policy distribution $\pi(a|s)$ as follows:

$$\forall s,a \;\; \pi(a|s) = \max\left(\frac{q_{sa}}{\alpha} - \tau(s), 0\right).$$

where $\tau(s)$ indicates $\tau(\frac{q_s}{\alpha})$.

The Lagrangian multiplier $c_s$ can be found from $\pi$ as follows:

$$
\begin{aligned}
\frac{c_s}{\alpha} &= \frac{1}{2} + \frac{1}{2} \sum_{a'} \left( \pi(a'|s) \right)^2 + \tau(s) \\
&= \frac{1}{2} + \frac{1}{2} \sum_{a' \in S(s)} \left( \frac{q_{sa'}}{\alpha} - \tau(s) \right)^2 + \tau(s) \\
&= \frac{1}{2} + \frac{1}{2} \sum_{a' \in S(s)} \left( \frac{q_{sa'}}{\alpha} \right)^2 - \sum_{a' \in S(s)} \frac{q_{sa'}}{\alpha} \tau(s) + \frac{K}{2} \tau(s)^2 + \tau(s) \\
&= \frac{1}{2} + \frac{1}{2} \sum_{a' \in S(s)} \left( \frac{q_{sa'}}{\alpha} \right)^2 - K \frac{\sum_{a' \in S(s)} \frac{q_{sa'}}{\alpha} - 1}{K} \tau(s) + \frac{K}{2} \tau(s)^2 \\
&= \frac{1}{2} + \frac{1}{2} \sum_{a' \in S(s)} \left( \frac{q_{sa'}}{\alpha} \right)^2 - \frac{K}{2} \tau(s)^2 \\
c_s &= \alpha \left[ \frac{1}{2} \sum_{a \in S(s)} \left( \left( \frac{q_{sa}}{\alpha} \right)^2 - \tau \left( \frac{q_s}{\alpha} \right)^2 \right) + \frac{1}{2} \right].
\end{aligned}
$$

To summarize, we obtain the optimality condition of (2) as follows:

$$
q_{sa} \triangleq \theta^\mathsf{T} \phi(s,a) + \gamma \sum_{s'} c_{s'} T(s'|s,a),
$$

$$
c_s = \alpha \left[ \frac{1}{2} \sum_{a \in S(s)} \left( \left( \frac{q_{sa}}{\alpha} \right)^2 - \tau \left( \frac{q_{s\cdot}}{\alpha} \right)^2 \right) + \frac{1}{2} \right],
$$

$$
\pi(a|s) = \max \left( \frac{q_{sa}}{\alpha} - \tau \left( \frac{q_{s\cdot}}{\alpha} \right), 0 \right).
$$

$\square$

## 1.4 Interpretation as Robust Bayes

In this section, the connection between MCTE estimation and a minimax game between a decision maker and the nature is explained. We prove that the proposed MCTE problem is equivalent to a minimax game with the Brier score.

**Theorem 5.** *The maximum causal Tsallis entropy distribution minimizes the worst case prediction Brier score, i.e.,*

$$
\min_{\pi \in \Pi} \max_{\tilde{\pi} \in \Pi} \; \mathbb{E}_{\tilde{\pi}} \left[ \sum_{a'} \frac{1}{2} \left( \mathbb{1}_{\{a'=a\}} - \pi(a'|s) \right)^2 \right] \quad \text{subject to} \quad \mathbb{E}_\pi \left[ \phi(s,a) \right] = \mathbb{E}_{\pi_E} \left[ \phi(s,a) \right] \tag{6}
$$

*where $B(s,a) = \sum_{a'} \frac{1}{2} \left( \mathbb{1}_{\{a'=a\}} - \pi(a'|s) \right)^2$ is the Brier score.*

*Proof.* The objective function can be reformulated as

$$
\begin{aligned}
\mathbb{E}_{\tilde{\pi}} \left[ \sum_{a'} \frac{1}{2} \left( \mathbb{1}_{\{a'=a\}} - \pi(a'|s) \right)^2 \right] &= \mathbb{E}_{\tilde{\pi}} \left[ B(s,a) \right] = \sum_{s,a} \rho_{\tilde{\pi}}(s,a) B(s,a) \\
&= \frac{1}{2} \sum_{s,a} \rho_{\tilde{\pi}}(s,a) \left( 1 - 2\pi(a|s) + \sum_{a'} \pi(a'|s)^2 \right),
\end{aligned}
$$

Hence, the objective function is quadratic with respect to $\pi(a|s)$ and is linear with respect to $\rho_{\tilde{\pi}}(s,a)$. By using the one-to-one correspondence between $\tilde{\pi}$ and $\rho_{\tilde{\pi}}$, we change the variable of inner maximization into the state action visitation. After changing the optimization variable, by using the

minimax theorem [4], the minimization and maximization of the problem (6) are interchangeable as follows:

$$\min_{\pi \in \Pi} \max_{\rho_{\tilde{\pi}} \in \mathbf{M}} \mathbb{E}_{\tilde{\pi}} \left[ \sum_{a'} \frac{1}{2} \left( \mathbb{1}_{\{a'=a\}} - \pi(a|s) \right)^2 \right]$$

$$= \max_{\rho_{\tilde{\pi}} \in \mathbf{M}} \min_{\pi \in \Pi} \mathbb{E}_{\tilde{\pi}} \left[ \sum_{a'} \frac{1}{2} \left( \mathbb{1}_{\{a'=a\}} - \pi(a|s) \right)^2 \right]$$

where sum-to-one, positivity, and Bellman flow constraints are omitted here. After converting the problem, an optimal solution of the inner minimization with respect to $\pi$ is easily computed as $\pi = \tilde{\pi}$ using $\nabla_{\pi(a''|s'')} \mathbb{E}_{\tilde{\pi}} [B(s,a)] = 0$. After applying $\pi = \tilde{\pi}$ and recovering the variables from $\rho_{\tilde{\pi}}$ to $\tilde{\pi}$, the problem (6) is converted into

$$\max_{\tilde{\pi} \in \Pi} \frac{1}{2} \sum_s \rho_{\tilde{\pi}}(s) \left( 1 - \sum_a \tilde{\pi}(a|s)^2 \right) = \max_{\tilde{\pi} \in \Pi} W(\tilde{\pi}),$$

where $\rho_{\tilde{\pi}}(s) = \sum_a \rho_{\tilde{\pi}}(s,a)$. Hence, the problem (6) is equivalent to the maximum causal Tsallis entropy problem. $\square$

In summary, the policy found in the maximum causal Tsallis entropy problem can be interpreted as the optimal decision maker considering the worst nature in sense of the Brier score.

## 1.5 Generative Adversarial Setting with Maximum Causal Tsallis Entropy

In this section, we convert the maximum causal Tsallis entropy problem (3) into the generative adversarial setting by adding a reward regularization defined as follows:

$$\max_{\theta} \min_{\pi} \quad -\alpha W(\pi) - \mathbb{E}_{\pi} [\theta^\mathsf{T} \phi(s,a)] + \mathbb{E}_{\pi_E} [\theta^\mathsf{T} \phi(s,a)] - \psi(\theta)$$

$$\text{subject to} \quad \forall s, a \quad \sum_{a'} \pi(a'|s) = 1, \quad \pi(a|s) \geq 0 \tag{7}$$

The proof consists of two parts. We first show that the maximization and minimization of the problem (7) are interchangable, which means that the solution of the maxi-min problem is equivalent to that of the mini-max problem.

**Theorem 6.** *The maximum causal Tsallis entropy problem (7) is equivalent to the following problem:*

$$\min_{\pi} \quad \psi^* \left( \mathbb{E}_{\pi} [\phi(s,a)] - \mathbb{E}_{\pi_E} [\phi(s,a)] \right) - \alpha W(\pi)$$

$$\text{subject to} \quad \forall s, a \quad \sum_{a'} \pi(a'|s) = 1, \quad \pi(a|s) \geq 0$$

*where $\psi^*(x) = \sup_y \{y^\mathsf{T} x - \psi(y)\}$.*

*Proof.* We first change the variable from $\pi$ to $\rho$ as follows:

$$\max_{\theta} \min_{\rho} \quad -\alpha \bar{W}(\rho) - \theta^\mathsf{T} \sum_{s,a} \rho(s,a)\phi(s,a) - \theta^\mathsf{T} \sum_{s,a} \rho_E(s,a)\phi(s,a) - \psi(\theta)$$

$$\text{subject to} \quad \forall s, a, \sum_{s,a} \rho(s,a)\phi(s,a) = \sum_{s,a} \rho_E(s,a)\phi(s,a),$$

$$\rho(s,a) \geq 0, \quad \sum_a \rho(s,a) = d(s) + \gamma \sum_{s',a'} T(s|s',a')\rho(s',a'), \tag{8}$$

where $\rho_E$ is $\rho_{\pi_E}$. Let

$$\bar{L}(\rho, \theta) \triangleq -\alpha \bar{W}(\rho) - \psi(\theta) - \theta^\mathsf{T} \sum_{s,a} \rho(s,a)\phi(s,a) + \theta^\mathsf{T} \sum_{s,a} \rho_E(s,a)\phi(s,a). \tag{9}$$

From Theorem 3, $\bar{W}(\rho)$ is a concave function with respect to $\rho$ for a fixed $\theta$. Hence, $\bar{L}(\rho, \theta)$ is also a concave function with respect to $\rho$ for a fixed $\theta$. From the convexity of $\psi$, $\bar{L}(\rho, \theta)$ is a convex

function with respect to $\theta$ for a fixed $\rho$. Furthermore, the domain of $\rho$ is compact and convex and the domain of $\theta$ is convex. Based on this property of $\bar{L}(\rho, \theta)$, we can use minimax duality [4]:

$$\max_{\theta} \min_{\rho} \ \bar{L}(\rho, \theta) = \min_{\rho} \max_{\theta} \ \bar{L}(\rho, \theta).$$

Hence, the maximization and minimization are interchangable. By using this fact, we have:

$$\max_{\theta} \min_{\rho} \ \bar{L}(\rho, \theta) = \min_{\rho} \max_{\theta} \ \bar{L}(\rho, \theta)$$

$$= \min_{\rho} \ -\alpha \bar{W}(\rho) + \max_{\theta} \left( -\psi(\theta) + \theta^{\mathsf{T}} \sum_{s,a} \left( \rho(s, a) - \rho_E(s, a) \right) \phi(s, a) \right)$$

$$= \min_{\rho} \ -\alpha \bar{W}(\rho) + \psi^* \left( \sum_{s,a} \left( \rho(s, a) - \rho_E(s, a) \right) \phi(s, a) \right)$$

$$= \min_{\pi} \ \psi^* \left( \mathbb{E}_{\pi} \left[ \phi(s, a) \right] - \mathbb{E}_{\pi_E} \left[ \phi(s, a) \right] \right) - \alpha W(\pi)$$

$\square$

### 1.6 Tsallis Entropy of a Mixture of Gaussians

The Tsallis entropy of a mixture of Gaussian distribution has an analytic form as follows:

**Theorem 7.** *Let* $\pi(a|s) = \sum_{i}^{K} w_i(s)\mathcal{N}(a; \mu_i(s), \Sigma_i(s))$. *Then,*

$$W(\pi) = \frac{1}{2} \sum_{s} \rho_{\pi}(s) \left( 1 - \sum_{i}^{K} \sum_{j}^{K} w_i(s)w_j(s)\mathcal{N}\left( \mu_i(s); \mu_j(s), \Sigma_i(s) + \Sigma_j(s) \right) \right), \qquad (10)$$

*where* $\mathcal{N}(x; \mu, \Sigma)$ *indicates a multivariate Gaussian density at point* $x$ *with mean* $\mu$ *and covariance matrix* $\Sigma$

*Proof.* The causal Tsallis entropy of a mixture of Gaussian distribution can be obtained as follows:

$$W(\pi) = \frac{1}{2} \sum_{s} \rho_{\pi}(s) \left( 1 - \int_{\mathcal{A}} \pi(a|s)^2 \mathbf{d}a \right)$$

$$= \frac{1}{2} \sum_{s} \rho_{\pi}(s) \left( 1 - \int_{\mathcal{A}} \left( \sum_{i}^{K} w_i(s)\mathcal{N}\left( a; \mu_i(s), \Sigma_i(s) \right) \right)^2 \mathbf{d}a \right)$$

$$= \frac{1}{2} \sum_{s} \rho_{\pi}(s) \left( 1 - \sum_{i}^{K} \sum_{j}^{K} w_i(s)w_j(s) \int_{\mathcal{A}} \mathcal{N}\left( a; \mu_i(s), \Sigma_i(s) \right) \mathcal{N}\left( a; \mu_j(s), \Sigma_j(s) \right) \mathbf{d}a \right) \qquad (11)$$

$$= \frac{1}{2} \sum_{s} \rho_{\pi}(s) \left( 1 - \sum_{i}^{K} \sum_{j}^{K} w_i(s)w_j(s)\mathcal{N}\left( \mu_i(s); \mu_j(s), \Sigma_i(s) + \Sigma_j(s) \right) \right)$$

$\square$

## 2 Causal Entropy Approximation

In our implementation of maximum causal Tsallis entropy imitation learning (MCTEIL), we approximate $W(\pi)$ using sampled trajectories as follows:

$$W(\pi) = \mathbb{E}_{\pi} \left[ \frac{1}{2} \left( 1 - \pi(a|s) \right) \right] \approx \frac{1}{N} \sum_{i=0}^{N} \sum_{t=0}^{T_i} \frac{\gamma^t}{2} \left( 1 - \int_{\mathcal{A}} \pi(a|s_{i,t})^2 \mathbf{d}a \right), \qquad (12)$$

where $\{(s_{i,t}, a_{i,t})_{t=0}^{T_i}\}_{i=0}^{N}$ are $N$ trajectories and $T_i$ is the length of the $i$th trajectory. Since the integral part of (12) is analytically computed by Theorem 7, there is no additional computational cost.

We have also tested the following approximation:

$$W(\pi) = \mathbb{E}_\pi \left[ \frac{1}{2} \left( 1 - \pi(a|s) \right) \right] \approx \frac{1}{N} \sum_{i=0}^{N} \sum_{t=0}^{T_i} \frac{\gamma^t}{2} \left( 1 - \pi(a_{i,t}|s_{i,t}) \right).$$

However, this approximation has performed poorly compared to (12).

For soft GAIL, $H(\pi)$ is approximated as the sum of discounted likelihoods

$$H(\pi) = \mathbb{E}_\pi \left[ - \log \left( \pi(a|s) \right) \right] \approx \frac{1}{N} \sum_{i=0}^{N} \sum_{t=0}^{T_i} -\gamma^t \log \left( \pi(a_{i,t}|s_{i,t}) \right).$$

Note that the same approximation (12) of $W(\pi)$ is not available for $H(\pi)$ since $-\int_{\mathcal{A}} \pi(a|s) \log \left( \pi(a|s) \right) \mathrm{d}a$ is intractable when we model $\pi(a|s)$ as a mixture of Gaussians.

## 3    Additional Experimental Results

In the multi-goal environment, the experimental results with other hyperparameters are shown in Figure 1.

## Footnotes

[1] $\sum_i\mu_i F(x_i)\leq F(\sum_i\mu_i x_i)$, for some $(x_1,\ldots,x_n)$ and $(\mu_i,\ldots,\mu_n)$ such that $\mu_i\geq 0$ and $\sum_i\mu_i=1$.

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

(a) Average Return

(b) Reachability

(c) Average Return

(d) Reachability

(e) Average Return

(f) Reachability

Figure 1: (a) and (b) show the average return and reachability of MCTEIL, respectively. (c) and (d) show the average return and reachability of soft GAIL, respectively. (e) and (f) show the average return and reachability of info GAIL, respectively. $k$ indicates the number of mixtures, $\alpha$ indicates an entropy regularization coefficient, and $c$ indicates a dimension of the latent code of Info GAIL.