[Reviews · NeurIPS 2018]

Reviewer 1



************************* Summary: ************************* This work introduces a new variant of imitation learning, MCTEIL, that replaces the common approach of maximizing the causal entropy within recent imitation learning approaches with maximizing the causal Tsallis entropy. This allows learning unimodal and multi-modal policies from demonstrations. The work is trained in an adversarial way, similar to GAIL [1], and is compared to behavior cloning, GAIL, and a baseline that is introduced in the paper, soft GAIL. ************************* Quality: ************************* The approach appears sound, although I did not check the math too heavily. The approach is well-supported with several theoretical proofs. There is no discussion about the weaknesses of the approach. ************************* Clarity: ************************* The paper is well-written and there were few typos. One disadvantage is that there were no hyper-parameters or architectural details included for the experiments, and so it is unlikely that the results can be reproduced. However, it is useful that the pseudo-code was provided. ************************* Originality: ************************* The main contribution to this work is that it describes how to incorporate causal Tsallis entropy into GAIL. This seems like an interesting formulation, however, it is unclear because this paper does not have a related works section. There are a few relevant works mentioned throughout the paper, but they either 1) are directly relevant to describing the background or 2) only mentioned briefly to state why they are not used as baselines. As such, it is unclear how much this work differs from other papers. For example, there are a couple works that were published in NIPS last year, InfoGAIL [2], and Multi-modal imitation learning [3], that learn multi-modal behaviors from demonstrations. The paper states that there are no comparisons with InfoGAIL because it uses a single Gaussian for the policy. However, InfoGAIL uses latent codes to capture multiple modalities. Additionally, the paper claims the latter work was excluded because it focuses on multi-task settings, but the experiments in that paper include both single and multi-task results. As such, it appears that these works should have been included in the comparisons, or at least warranted further discussions. ************************* Significance: ************************* It is useful to learn multi-modal policies, in particular because they provide alternatives to difficult policies to achieve, and additionally can highlight different preferences among experts. This paper has nice theoretical foundations, and the pseudo-code is pretty clear, so it would be easy to make a couple of adjustments to GAIL if researchers are interested in using MCTEIL. However, because this paper does not compare against relevant works, it is difficult to assess how significant it is. Additionally, the paper needs more analysis of the results. Why does MCTEIL outperform in Half-Cheetah and Reacher, but not the other environments? Furthermore, there were only 3 averaged runs, so it is unclear how significant the results are. ************************* Typos ************************* Line 39: modeling expert’s -> modeling the expert’s Line 109: to the expected -> with the expected ************************* References: ************************* [1] Ho, Jonathan, and Stefano Ermon. "Generative adversarial imitation learning." Advances in Neural Information Processing Systems. 2016. [2] Li, Yunzhu, Jiaming Song, and Stefano Ermon. "InfoGAIL: Interpretable Imitation Learning from Visual Demonstrations." arXiv preprint arXiv:1703.08840 (2017). [3] Hausman, Karol, et al. "Multi-modal imitation learning from unstructured demonstrations using generative adversarial nets." Advances in Neural Information Processing Systems. 2017. Update: I have read through the reviews and the author's response. I still think it is necessary to have discussions in the paper relating the work to other relevant literature, but I am willing to change my review to weak accept, given the theoretical contributions and commitment to include the relevant works and experiments in the paper.

Reviewer 2



This paper develops a principle of maximum causal Tsallis entropy (MCTE) and applies it to imitation learning tasks. Overall, the paper makes a solid contribution to imitation learning in both theory and practice. I favor acceptance, though I have some concerns about the framing and derivation assumptions below. Compared to the principle of maximum causal entropy (MCE), it provides sparseness in its probability estimates. In part, this can be understood from eq. (8): MCE is a robust (causal) logloss minimizer, while MCTE is a robust Brier score minimizer. Since logloss penalizes (incorrect) sparsity infinitely, while the Brier score minimizes it finitely, sparse solutions are more easily permitted. Multi-modality is the other main benefit claimed by this approach. First, some of the phrasing (“from a uni-modal to multi-modal policy distribution”) should be clarified, e.g., to “from both uni-modal and multi-modal policy distributions” since it is unclear. Still, the multi-modality claim is hard for me to understand. It seems to be justified by the Tsallis entropy’s analytical expressibility in continuous domains when combined with mixture density networks (Theorem 7) and experimental results. However, this seems like more of a computational nicety than a fundamental difference between approaches. Can this be clarified? Does multi-modality have meaning in discrete action spaces? Are strictly Markovian policies implied by the Tsallis entropy? There is a noticeable difference in the built in Markovian assumptions of your formulation compared to maximum causal entropy. Line 109 already makes a Markovian assumption about the policy, but if instead the Tsallis entropy is defined over non-Markovian policies (i.e., by conditioning on all previous states and actions), does the solution reduce to a Markovian policy? This would not be a fundamental problem, but could have implications for changes in state-action space representations, and warrant some discussion. --------- Thank you for your clarifications. Adding some discussion of the Markov assumption and differences from the Shannon entropy in this regard would be much appreciated.

Reviewer 3



The paper extends maximum entropy-based approaches for imitation learning by replacing information entropy by Tsallis entropy. The motivation is that while the former approach leads to softmax distribution policies, the latter finds sparsemax distribution policies, allowing zero probabilities on bad actions. Although a direct extension, the approach is novel and well-motivated. I appreciated its interpretation with the Brier score. In the experiments, it would have been interesting to evaluate the case where k is misspecified in the multi-goal environment. How was k chosen for the other domains? The exposition could have been a little bit more self-contained and clearer. For instance, the definition of a sparse distribution could be recalled. Minor remarks and typos: l.70: the optimal policy distribution -> an optimal policy distribution (there's no unicity) l.79: casual l.112: that that Th.1 is an "old" result, it can already be found in [Puterman, 1994] l.137: \gamma is missing (11): \rho_\pi(s) is not defined l.258: is an \alpha missing?